# Field Crop Evaluation of Polymeric Nanoparticles of Garlic Extract–Chitosan as Biostimulant Seed Nano-Priming in Cereals and Transcriptomic Insights

**DOI:** 10.3390/polym16233385

**Published:** 2024-11-30

**Authors:** María Mondéjar-López, Alberto José López-Jiménez, Lourdes Gómez-Gómez, Oussama Ahrazem, Joaquín Calixto García-Martínez, Enrique Niza

**Affiliations:** 1Instituto Botánico, Departamento de Ciencia y Tecnología Agroforestal y Genética, Universidad de Castilla-La Mancha, Campus Universitario s/n, 02071 Albacete, Spain; maria.mondejar@uclm.es (M.M.-L.); albertojose.lopez@uclm.es (A.J.L.-J.); marialourdes.gomez@uclm.es (L.G.-G.); oussama.ahrazem@uclm.es (O.A.); 2Escuela Técnica Superior de Ingeniería Agronómica y de Montes y Biotecnología, Departamento de Ciencia y Tecnología Agroforestal y Genética, Universidad de Castilla-La Mancha, Campus Universitario s/n, 02071 Albacete, Spain; 3Facultad de Farmacia, Departamento de Ciencia y Tecnología Agroforestal y Genética, Universidad de Castilla-La Mancha, C/José María Sánchez Ibáñez s/n, 02008 Albacete, Spain; 4Department of Inorganic and Organic Chemistry and Biochemistry, Faculty of Pharmacy, Universidad de Castilla-La Mancha, C/José María Sánchez Ibáñez s/n, 02008 Albacete, Spain; 5Instituto de Biomedicina (IB-UCLM), Universidad de Castilla-La Mancha, C/Almansa 13, 02008 Albacete, Spain

**Keywords:** crop management, garlic extract, chitosan nanoparticles, biostimulant properties, transcriptomic analysis

## Abstract

Current crop management worldwide is shifting toward the use of environmentally friendly products. With this objective, we developed a new phytosanitary product with biostimulant properties based on the encapsulation of garlic extract at a lower dose (<0.1%) in chitosan nanoparticles as a seed nano-priming agent. In the present study, the morphology of the nanoparticles, their stability under prolonged storage conditions, and their efficacy as a biostimulant are evaluated on cereals in rainfed crops, and the activities were correlated with a transcriptomic analysis. The nanoparticles showed a spherical shape and had a maximum size close to 200 nm with satisfactory stability at 4 °C, reducing the probability of aggregation processes in the nanoparticles. The biostimulant properties of the nano-priming agent were evaluated in a field experiment with wheat, barley, and oat seeds at 30 and 90 days, showing that plants treated with nanoparticles showed significant differences with higher values in root development, leaf length, and total plant weight. Finally, through a RNA-SEQ analysis of the treated wheat seeds, we have confirmed that the nano-treatment showed a higher increases in regard to development, metabolism, and plant response genes compared with untreated seeds.

## 1. Introduction

The climate emergency, together with the increase in resistance to fungal species, population growth and toxicities generated in animals [1] and humans by the indirect consumption of conventional treatments such as triazoles, has led governments to develop new regulatory frameworks to reduce the use of many current crop protection products and to ban them in coming years [2]. However, the search for novel and efficient products that achieve the desired results in crop management remains challenging.

Nanotechnology has emerged as an efficient alternative that can help overcome this problem through the use of many nanoscale raw materials that can serve as a platform to encapsulate, protect, deliver, and increase the effect of many active ingredients to increase and facilitate crop management [3,4]. Nano-agri or agri-nanotechnology is a recent term applied to a novel field that focuses on crop management with the use of different nanomaterials [5,6]. The use of nano-agri approaches has many advantages, including increased sustainability and cost and time efficiency, whereas disadvantages such as potential toxicity and the environmental damage of different inorganic nanomaterials still need to be investigated [7]. However, green nanotechnology can offer the safe use of nanotechnology in crop management through the use of many raw materials such as polysaccharides, natural biopolymers, lipids, and gums, which are widely studied as co-formulants or basic substances in several formulations of current plant-protection products [8,9]. Chitosan is a natural biopolysaccharide widely used in agriculture; it is registered as a basic substance by the European science, safe food, and sustainability authority (EFSA) and is the most used ingredient in ecological agriculture due to its properties as an antifungal, antibacterial, and elicitor agent [10,11].

Nano-priming is a novel approach to seed treatment technologies that involves the application of different nanomaterials or nano-formulations on seeds before sowing [12]. Seed nano-preparation has emerged as a future application of seed technologies due to its many advantages over convective methods of seed preparation. The use of different nanomaterials can form nanopores that facilitate water uptake, activate the reactive oxygen species (ROS)/antioxidant mechanism in seeds, and increase hydroxyl radicals to loosen cell walls, inducing the rapid hydrolysis of starch by stimulating amylase and increasing seed germination [13]. In addition, increased levels of mild ROS at germination stages act as primary signaling cascade events involved in secondary metabolite production and stress tolerance [14].

Recently, chitosan nanoparticles have been proven to regulate plants positively at lower concentrations than chitosan [15]. Moreover, the adsorption of chitosan on the seed surface is also more effective than the adsorption of the compound itself. Nanoparticles made from this polysaccharide are able to stimulate the growth of wheat and other cereals by upregulating indole-3-acetic acid synthesis genes and downregulating metabolic genes. The additional uptaking of nutrients such as calcium, magnesium, potassium, phosphorus, and nitrogen occurs [16].

Plant metabolites, such as essential oils, terpenes, or extracts, are being recognized by a large number of research groups and companies due to their benefits in regard to efficient crop management [3]. Many of them, such as thymol, geraniol, eugenol, or garlic extract, are currently used as plant protection products due to their antifungal and nematicide activities [17]. Also, garlic extract has been shown to improve crop quality and soil conditions under biotic and abiotic stress conditions, showing noticeable differences in germination indices and seedling growth (particularly root growth and fresh weights) in primed seeds in a dose-dependent manner [18].

As seen in our previous works, we obtained different novel antifungal and seed-priming treatments using eco-friendly and nanotechnology approaches through circular economy processes [3,19]. Although the treatment showed encouraging preliminary results as an antifungal and elicitor product, as with all potential products, its activity should be evaluated under normal growing conditions in complete vegetative cycles in order to assess its behavior in circumstances that are as close as possible to reality.

On the other hand, scaling-up is a crucial step that any potential product or treatment must go through before it can be taken to the field. In this case, several mechanisms and instrumentations are already available in the industry for carrying out encapsulation processes in nanoparticles through similar techniques, as has already been observed in different fields such as medicine and cosmetics [20,21]. Based on these approaches, the company involved in this work developed different approaches to achieve the production of these nanoparticles in an efficient manner in order to obtain industrial-scale amounts of the product. One of the proposed products consists of the encapsulation of garlic extract at very low doses (<0.1%) in chitosan-modified nanoparticles to obtain new seed biostimulants or nano-priming-based products.

In an experimental rainfed system, we assessed the effectiveness of nanoparticles of Garlic Extract–Chitosan (GE-NPCH) on three cereal crops in this study: barley, oat, and durum wheat. The main aim of this study was to identify a substitute for tebuconazole (TB), which is used to protect seeds from fungal diseases and stimulate growth. The experimental analysis was performed between the first planting phase and the vegetative stage at 90 days, which is the usual active time for pre-emergence treatments. The goal of this thorough analysis was to determine whether GE-NPCH may improve crop yield and resilience in rainfed environments, providing a workable and sustainable substitute for traditional chemical treatments.

## 2. Materials and Methods

### 2.1. Materials

Nanoparticles of GE-NPCH were provided by Candelo biotech SL (Albacete, Spain) using Chitosan with 190–310 Kda and ≥75% of deacetylation degree. Seeds were obtained from a local seed supplier. All reagents were supplied by Merck (Barcelona, Spain).

### 2.2. Morphological Studies

Morphological surface and shape analyses of NPCH and GE-NPCH were performed using scanning electron microscopy (SEM) and transmission electron microscopy (TEM). Samples were sputtered with Pt and observed using a Jeol (Tokio, Japan) 7800F electron microscope at 20 kV and a Jeol JEM 210 TEM microscope operating at 200 kV, equipped with an Oxford Link EDS detector.

### 2.3. Stability Studies

Dynamic light scattering (DLS), also known as photon correlation spectroscopy, and zeta potential (ZP) techniques were used to investigate the hydrodynamic size and surface charge of nanoparticles, respectively [19]. The particle characterization of nano-formulations (size, ZP, and polydispersity index (PDI)) was determined using a Zetasizer (3000HSM Malvern Ltd., IESMAT, Madrid, Spain) with the following specifications: a chitosan refractive index (RI) of 1.700, an absorption index of 0.010, and a water solvent RI of 1.33 with a viscosity of 0.8872 cP. Measurements were performed in triplicate. To perform a stability test, the nanoparticle formulation was stored at 4 °C for 11 months. Size, PDI, and Z-potential measurements were taken at 2, 5, 10, and 11 months after the beginning of the experiment.

### 2.4. Evaluation of GE-NPCH as a Seed Coating Agent in a Rainfed Crop Experiment

To evaluate the morphological effects on the plants, batches of 3 kg each of wheat, barley, and oat seeds were spray-treated with GE-NPCH at a concentration of 10 mg/mL based on effects evaluated in previous studies. The effect was compared to the concentration recommended by the manufacturer of the TB formulation (45 mg/mL of product with 9 mg/mL of pure TB) and to the same seeds treated with commercial TB using our coating system (sprayer) to evaluate the deficiencies of the coating system compared to seeds coated by a local seed company (Cereales Candelo SL, Albacete, Spain). Untreated seeds were also used in a multi-comparative study. Seeds were grown in Albacete (Spain) at the coordinates 39°07′06.2″ N 1°30′40.8″ W in an experimental plot of 27,000 m^2^ under rainfed conditions with an accumulated rainfall of 62.2 L/m^2^. They were divided into microplots of 3000 m^2^ for each treatment, as shown in Figure 1 of this section. No fertilizer or any other treatment was used during the crop cycle to avoid altering the effect of the treatments in the first 3 months of cereal cultivation. Plant samples were collected after 30 and 90 days of treatment. Plants were evaluated by measuring their weight, root length, leaf length in cm, and other morphological parameters. Approximately 20 plants were collected, of which 10 were homogeneously selected based on the most representative morphology of the sample population for evaluation.

### 2.5. Flavonoid, Polyphenol, and Chlorophyll Content in Treated Plants

We evaluated the levels of polyphenols, flavonoids, and chlorophylls to determine whether the different treatments caused changes in cellular activity and plant physiology. Because polyphenols and flavonoids are involved in defensive mechanisms against infections and environmental stresses, they were assessed as markers of plant antioxidant capacity and stress response.

#### 2.5.1. Determination of Total Polyphenol Content

To determine the total amount of polyphenols in the aqueous extract, the Folin–Ciocalteau method was carried out [22]. Briefly, 0.1 mL of aqueous extract was mixed with 2 mL of 2% Na_2_CO_3_, 2.8 mL of H_2_O, and 0.1 mL of Folin–Ciocalteau reagent. After mixing, the color change was measured by absorbance at 750 nm. Gallic acid (GAE) was used as a standard at different concentrations (10–200 ppm). The experiment was carried out with *n* = 10.

#### 2.5.2. Determination of Total Flavonoid Content

Flavonoid content in the aqueous extract was determined by a colorimetric method using AlCl_3_·6H_2_O [22]. Briefly, 0.5 mL of the aqueous extract was mixed with 1.5 mL of ethanol, 0.1 mL of 10% AlCl_3_·6H_2_O, 0.1 mL of 1 M CH_3_COOK, and 2.8 mL of H_2_O. After mixing, the color change was evaluated at 415 nm. Quercetin (QE) was used as a standard at different concentrations (8–500 ppm). The experiment was carried out with *n* = 10.

#### 2.5.3. Determination of Chlorophyll Content

The total chlorophyll content in the leaves of treated seed plants was assessed as described by [21] Asimovic et al. (2016) with some modifications. Specifically, 50 mg of powdered leaves were extracted with 300 µL of 80% acetone. The samples were then centrifuged at 10,000× *g* for 10 min, and the absorbance of the supernatant was measured at 662 and 644 nm. The experiment was carried out with *n* = 10.

### 2.6. Transcriptome Analysis

#### 2.6.1. RNA Extraction, Library Preparation, and Sequencing

Total RNA was extracted from wheat seeds using the TRIzol™ Reagent (ZYMO Research, Irvine, CA, USA) according to the manufacturer’s instructions and then treated with RNase-free DNase I to remove any residual genomic DNA [22]). RNA samples were subjected to mRNA enrichment using the TruSeq Stranded mRNA Library Prep Kit (Illumina, San Diego, CA, USA) following the manufacturer’s protocol. Libraries, configured for 150 bp paired-end sequencing, were constructed using the TruSeq Stranded mRNA Library Prep Kit (Illumina, USA) and subsequently sequenced on the Illumina NovaSeq6000 platform.

#### 2.6.2. Data Processing and Analysis

The RNA-seq raw read quality was assessed using FastQC. To avoid biases in the analysis, adaptor sequencing artifacts and low-quality fragments were subsequently removed using Trimmomatic (v0.39) and the remaining reads were aligned to the reference IWGSC1.1 version of the wheat genome/transcriptome using the alignment software RNA STAR implemented in Galaxy (Galaxy Version 2.7.11a+galaxy0). Transcript abundance estimation and a differential expression analysis were performed using featureCounts and DESeq2, respectively. A functional enrichment analysis of differentially expressed genes was conducted using Gene Ontology (GO) and the Kyoto Encyclopedia of Genes and Genomes (KEGG) pathways through the iDEP 2.0 web application.

### 2.7. Statistics

The obtained data were statistically analyzed using one-way ANOVA and Dunnett’s Multiple Comparisons test with the statistical software GraphPad Prism version 5.0.0 for Windows (GraphPad Software, San Diego, CA, USA). Differences were considered significant at *p* < 0.05 (95% probability level).

## 3. Results and Discussion

### 3.1. Morphology Characterization of GE-NPCH

Scanning electron microscopy (SEM) is a commonly used technique to analyze the size distribution, surface roughness, and various morphological features of nanomaterials or nano-formulations [23]. As shown in Figure 2a, the nanoparticles provided by Candelo biotech SL displayed a homogeneous size distribution with average sizes close to 80 nm and 200 nm and a spherical shape typical of biopolymeric nanoparticles such as alginate [24] and chitosan nanoparticles [19,25]. Previously developed lab-scale formulations of garlic essential oil encapsulated in chitosan nanoparticles showed a spherical shape and homogeneous distribution, achieving a larger size (400 nm) than the nanoparticles observed in the present industrial-scale formulation evaluated by the aforementioned company [19,25].

### 3.2. Stability Properties of GE-NPCH

To determine the stability of the nanoparticle formulation containing garlic extract, the DLS technique was used. The homogeneity of the samples was practically unaffected during the 11-month period, with only a slight increase in the PDI. Additionally, the zeta potential did not change significantly over the storage time, with only a minor decrease being observed. All measurements conducted during the stability test showed a positive surface charge in the range of +41.8 to +43.7 mV, which is consistent with previous studies on chitosan nanoparticles encapsulating different active molecules (Table 1). These positive charges arise from the protonation of the amino groups.

Zeta potential is a crucial factor for nanoparticle stability in aqueous media, with values greater than +30 mV and less than −30 mV providing sufficient electrical repulsion to prevent nanoparticle aggregation, thus ensuring a more stable nanoparticle suspension [24] (Bhattacharjee, 2016). Regarding particle size, the radius increased over time (from 251.1 to 339.0 nm), with an increase of 4.8% at the second month, 19.4% at 5 months, 23.4% at 10 months, and 35.0% at 11 months. This size variation over time could be attributed to the swelling process. Similar results were obtained by., who reported that, after 12 months, the size of nanoparticles prepared by ionotropic gelation remained similar to that of freshly prepared samples at both 4 °C and 25 °C [26]. In another study investigating the stability of chitosan nanoparticles over 1 month, size changes of less than 10% were observed [27], which is consistent with our observed size variation of 4.8% over two months (Figure 2b).

### 3.3. Evaluation of Seed Treatment with GE-NPCH in Rainfed Crop

#### 3.3.1. Morphological Evaluation of Wheat

The images captured and shown in Figure 3a illustrate the differences observed using all treatments tested at 30 and 90 days after sowing. Morphologically, visual examination at 30 days showed that plants treated with GE-NPCH exhibited greater rooting and root length, reaching a larger size than the other seeds, with treated plants exceeding 8 cm in height, whereas untreated seeds reached a size of less than 4 cm.

These results were reflected in the statistical study carried out using Dunn’s multiple comparison test (Figure 4) to establish significant differences after treatments. The results show a significant increase in root length in seeds treated with GE-NPCH at 30 days compared to the rest of the treatments and a significant decrease in the same parameter in the case of TB1, suggesting insufficient seed coating compared to TB2.

Nevertheless, the differences increased significantly after 90 days, as observed in the right image of Figure 3, where the seeds treated with nanoparticles showed greater root and leaf length and higher tillering. These data are also seen in Figure 4, where the significant difference is confirmed, showing that the parameters of root length (11.4 cm), tillering number (6), total weight (9.5 g), and root weight (0.21 g) are significantly higher in treated plants than in TB and untreated plants, confirming the aforementioned pattern. Likewise, the seeds treated with GE-NPCH showed higher values of main shoot weight and number of roots compared with the other treatments, despite not showing significant differences in the other parameters after Dunnett’s multiple comparison study. In the early stages of development, vigor in plants treated with the new treatment was evident because, during these stages, plants require more resources to overcome various biotic and abiotic stresses [28,29].

#### 3.3.2. Morphological Evaluation of Barley

In contrast to the wheat samples, the GE-NPCH treatment on barley resulted in an increase in the development of the aerial parts of the plant 30 days after the beginning of cultivation, as illustrated in Figure 3b, where thicker and longer leaves are observed. In this image, a greater appearance of roots can be seen, although the differences in root length are not as pronounced compared to wheat.

Figure 3b shows that the lengths of the aerial and green parts of the leaves of the plants treated with GE-NPCH were greater after 30 days of cultivation, reaching a length of 10 cm for the green part, while the shortest length was 7 cm for TB1. However, as the statistical analysis shows, the most significant differences were found in the development of the green part of the leaf, where a greater length was observed in the case of plants treated with nanoparticles.

Although a greater increase in leaf growth was observed after 90 days, as shown in Figure 5, plants treated with the ecological nanoparticles exhibited a greater increase in root length, a greater number of roots, and a more robust consistency. Additionally, as observed in wheat, there was an increase in exponential tillering and the aerial part had a greater length.

In Figure 5, data evaluated using Dunn’s multiple comparison test indicated the trends seen in barley plants treated with GE-NPCH, where a significant increase in total plant weight and root weight was observed compared to the conventional tebuconazole treatment, mirroring the trend observed in wheat. On the other hand, very significant differences were observed in the case of TB1 in root weight and number, displaying a decrease in root length. TB2 confirmed the phytotoxicity effect in root development observed at the tested dose at 30 and 90 days, as several studies confirm that tebuconazole is likely to feature phytotoxicity and slow down plant growth when applied excessively during leaf or seed treatment. This is due to the effects of triazole fungicides on gibberellin phytohormone biosynthesis, which inhibits seed germination and plant growth [30].

#### 3.3.3. Morphological Evaluation of Oat

Morphological and visual analyses of oat samples collected at 30 days showed significant differences between untreated and treated plants. As shown in Figure 3c, both the roots and aerial parts of plants treated with GE-NPCH had greater sizes, thicknesses, and numbers of roots. The results were more significant after 90 days of sowing when the nanoparticle-treated plants displayed an increase in the total aerial part and longer and more numerous roots, showing a clear invigorating effect.

Figure 3 shows that, after 30 days of cultivation, the length of the aerial part, root length, and total weight of the plants treated with GE-NPCH were greater. The superior development of the roots of plants treated with the ecological nanoparticles was evident in the Dunn’s multiple comparison test carried out on the samples obtained after 30 days of development. Figure 4 indicates a higher root growth and total weight of plants treated with GE-NPCH, with significant differences being seen compared to tebuconazole and untreated plants. However, as the statistical analysis shows, the most significant differences were found in the development of root length, where a greater root length was observed in plants treated with nanoparticles, with roots close to 9 cm compared to 5 cm in untreated seeds. On the other hand, tebuconazole treatment significantly differed in regard to leaf length, supporting the aforementioned phytotoxicity effect in barley, as observed in other studies [31].

Similarly to what was observed in the other species tested, differences with the ecological nanoparticle treatment compared to the other treatments increased significantly after 90 days. In oat plants, as shown in Figure 3, this trend was maintained, and very significant differences were observed in the morphological and visual evaluation, where greater root growth, a greater number of roots, more pronounced tillering, and a greater number of shoots were found, supporting the invigorating behavior of the new ecological treatment.

To confirm whether the dimensional and gravimetric differences evaluated during the visual test are significant in tangible data, a Dunn’s multiple comparison test was performed, as with the other species tested. The data observed in Figure 5 reflect what was observed in Figure 8, where a greater length and number of roots and a greater weight of the plants were observed, mainly associated with an increase in the root system of the plant, showing significantly higher values compared to untreated plants and those treated with the conventional treatment.

### 3.4. Physiological Evaluation in Plants

The analysis of these pigments provides valuable information on the physiological state of the plant, as the photosynthesis process is vital for the correct development of the plant. An increase or reduction in pigments in the different treatments indicates whether something beneficial or detrimental is happening to the plant [3,32].

Phenolic compounds are part of the secondary metabolism of many plants and are organic compounds whose molecular structures contain at least one phenol group, an aromatic ring linked to a hydroxyl group. These compounds are slightly acidic and mostly highly oxidizing. They can have many functions within plants (there can be up to 10,000 different types of phenolic compounds), including uses involving pollination, mechanical support, and photoprotection; however, their main role is defensive [33,34].

When faced with biotic stress, such as infection, plants secrete a battery of phenolic compounds that strongly oxidize the intracellular environment. This has two consequences: first, they may be able, by themselves, to eliminate the pathogen; second, they activate the plant’s defense genes, creating a much more aggressive environment for the pathogen [35,36].

The results shown in Table 2 represent the total contents of chlorophylls, polyphenols, and flavonoids extracted from leaves collected after 90 days. For wheat, the first column indicates the values obtained for chlorophylls, where no differences between treatments were observed. The values obtained for total polyphenol content show lower levels in the treatments than in the control, and the same trend is observed for flavonoid content. However, the differences do not reveal a significant alteration, maintaining the typical values usually reported at this stage of vegetative development, indicating that none of the treatments produced any phytotoxic effect or undesired alteration during the evaluated period [37,38].

To evaluate the safety of the barley profile treatment, measurements of the above metabolites were performed to assess whether the treatment influences the cellular activity involved in stress and growth. The results shown in Table 2 confirm the biosafety trend observed in the other cereal species, which exhibited behavior similar to the other treatments and untreated samples without a clear trend in the treatment used. Thus, the highest values for chlorophylls were found in untreated barley, while the highest values for polyphenols and flavonoids were found in TB2 and TB1, respectively. However, no clear trend or significant change was observed in the values between treatments.

For the oat treatments, Table 2 shows the same trend as the previous trials, with no signs of alteration in the physiological functioning of the plants treated with the nanoparticles compared to the rest of the treatments, showing similar values and confirming that it is a safe treatment for this plant. These values are similar for chlorophylls; however, for polyphenols, the value is slightly higher for TB1, while, for flavonoids, the highest value is for the nanoparticle treatment.

### 3.5. Transcriptomic Analysis of the Elicitor Effect of GE-NPCH on Germination

In our previous work [19], we characterized the effect of dressing with GE-NPCH (with a higher dose of garlic essential oil) on the germination of wheat, barley, and rye seeds compared with tebuconazole or empty chitosan nanoparticles. Although the germination percentages among the different treatments for each cereal were similar, wheat seeds treated with GE-NPCH exhibited morphological characteristics at 15 days with increased weight. Therefore, we assessed the early transcriptional changes induced by treatment in wheat seed germination to determine the potential molecular mechanism behind the elicitor effect on germination by analyzing gene expression using RNA-seq before and after 24 h of germination with or without GE-NPCH treatment.

In total, nine libraries were generated that produced a total of 100 million reads with quality scores greater than or equal to 30 (Q30) in more than 95% of cases. The percentage of reads mapping to the reference genome was higher than 66% in all cases, except for one library corresponding to the control treatment, which was discarded from further analysis.

To determine how GE-NPCH treatment alters gene expression during early germination, we first compared GE-NPCH treatment versus untreated seeds. As seen in the principal component analysis in Figure 6a, there was a clear separation between GE-NPCH and control along the principal component axes, establishing clear patterns of expression between the different groups studied, as shown in the heatmap displaying clusters (k-means) among the 2000 most variable genes in Figure 6c.

When we examined the differentially expressed genes (DEGs) of the control versus the GE-NPCH treatment, as shown in Figure 6, we found a clear overlap of DEGs between the control and GE-NPCH. The number of DEGs was higher in the GE-NPCH treatment vs. the time 0 comparison than in the control treatment comparison (10,102 and 6017, respectively). Thus, in the group of upregulated genes in the control treatment at 24 h of germination versus time 0, 4143 out of a total of 4564 DEGs (90.77%) were also upregulated in the comparison of GE-NPCH treatment versus time 0. Similarly, 1284 of 1450 (88.55%) of the genes downregulated in control vs. time 0 were downregulated in GE-NPCH vs. time 0. When we compared these two treatments (GE-NPCH vs. control), we found 1211 upregulated and 549 downregulated genes.

DEGs were analyzed for Gene Ontology (GO) and Kyoto Encyclopedia of Genes and Genomes (KEGG) enrichment to elucidate their functional roles and involved pathways. Figure 7 shows network plots of KEGG pathways and GO (Biological Process) based on the present analysis. In the case of KEGG pathways, alpha-linolenic acid metabolism, phenylpropanoid metabolism, oxidative phosphorylation, and metabolic pathways appear to be overexpressed among the differentially expressed genes. Conversely, RNA degradation pathways, the pentose phosphate pathway, glycerolipid metabolism, and related processes of mismatch repair, nucleotide excision repair, and DNA replication appear to be repressed. In relation to GO biological processes, two clear clusters of terms related to chromosomal organization and nucleosome assembly and glutathione metabolism are overexpressed. Finally, a cluster of genes related to the response to different stimuli, all of which are related to the response to abscisic acid, are repressed.

In both analyses, the upregulation of genes involved in the metabolism of glutathione (GSH) stands out (Figure 8). GSH has emerged as a significant signaling molecule that plays a crucial role in regulating ABA signal transduction and the developmental events linked with it [39]. The equilibrium between seed dormancy and germination is governed by a dynamic interplay between the synthesis and breakdown of abscisic acid (ABA) and gibberellins (GAs). Previous studies have demonstrated that GSH induces seed dormancy release in barley [40]. Elevated levels of GSH-containing glutathionylated proteins/compounds, such as GRX and GSNO, inhibit ABA signaling during seed germination, thereby suppressing ABA signaling and promoting seed germination [41,42].

## 4. Conclusions

In summary, we evaluated the long-term stability of the nanoparticles stored at 4 °C. The new seed-dressing treatment demonstrated successful biostimulant properties in wheat, showing a significant increase in root length, the number of shoots, and total weight relative to untreated and tebuconazole-treated seeds. Similar patterns were observed for oats, showing significant differences in root promotion, confirming their vigorous properties. Barley seeds treated with nanoparticles also showed a significant increase in root and leaf length compared to tebuconazole treatment, confirming enhanced plant development properties. The phytosafety profile of the nanoparticles was evaluated through the measurement of secondary metabolites, obtaining satisfactory amounts of polyphenols, chlorophyll, and flavonoids in the nano-treated plants. Additionally, RNA-seq studies on treated seeds showed a significant increase in the expression of genes involved in plant development, metabolism, and defense, confirming the full-spectrum protection of the nano-treatment due to its effect on genes involved in glutathione metabolism. Finally, we can confirm that the new seed nano-treatment, based on our previous works and scaled-up by Candelo biotech SL., offers a novel and promising approach to improving cereal crop management through significant biostimulant activity.

## Figures and Tables

**Figure 1 polymers-16-03385-f001:**
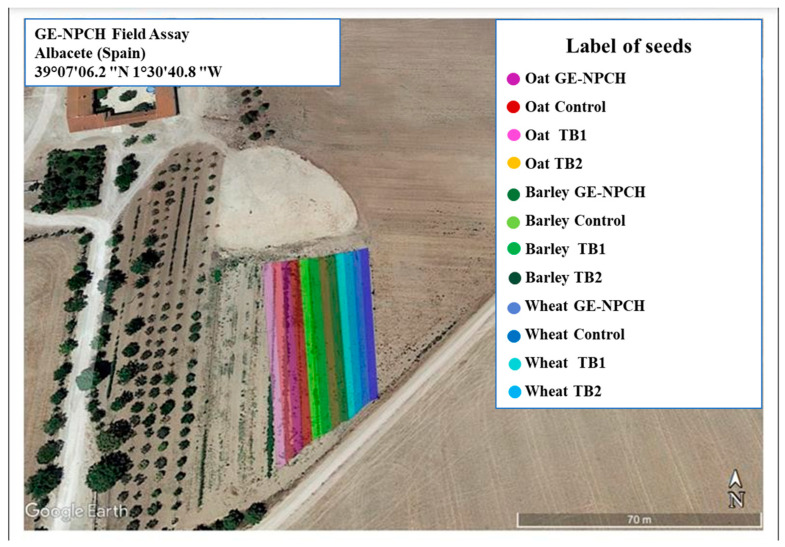
Seed labeling and field experiment localization.

**Figure 2 polymers-16-03385-f002:**
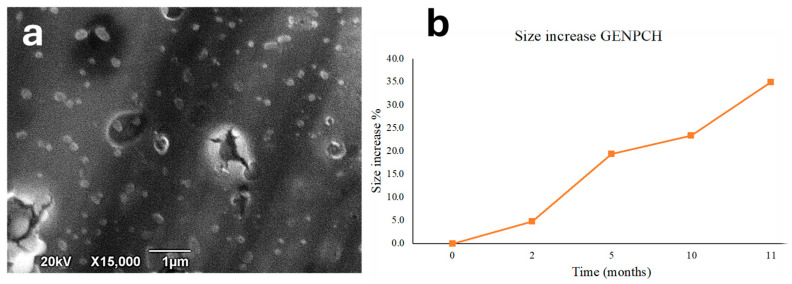
(**a**) SEM image of GE-NPCH. (**b**) Stability study of GE-NPCH.

**Figure 3 polymers-16-03385-f003:**
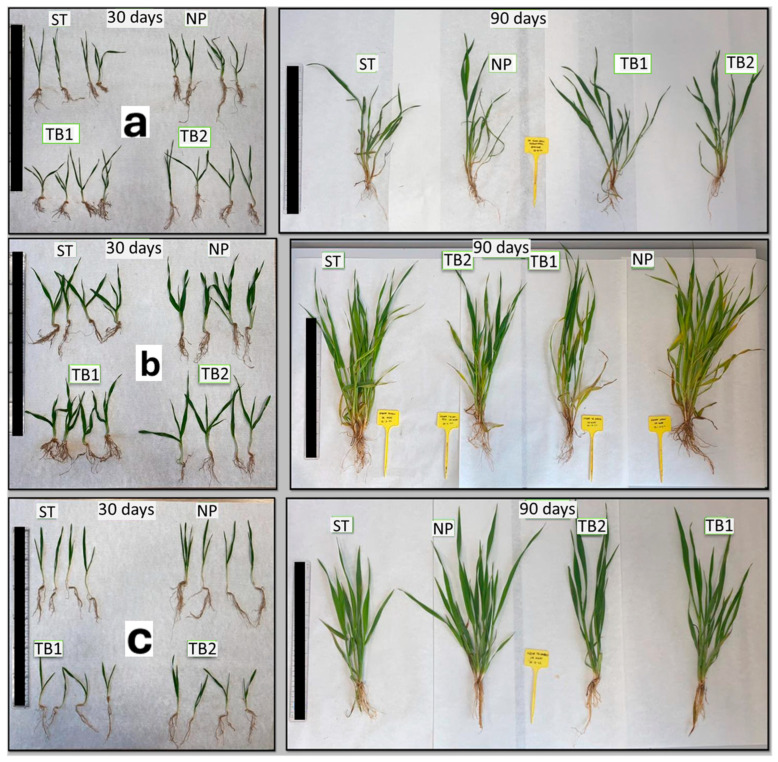
Morphological analysis of the different samples: (**a**) wheat, (**b**) barley, and (**c**) oat plants. On the left, samples were obtained 30 days after sowing; on the right, samples were obtained 90 days after sowing. C: control (untreated); NP: treated with GE-NPCH; TB1: treated by spraying with TB; TB2: seeds treated by the seed grower. Scale bar = 30 cm.

**Figure 4 polymers-16-03385-f004:**
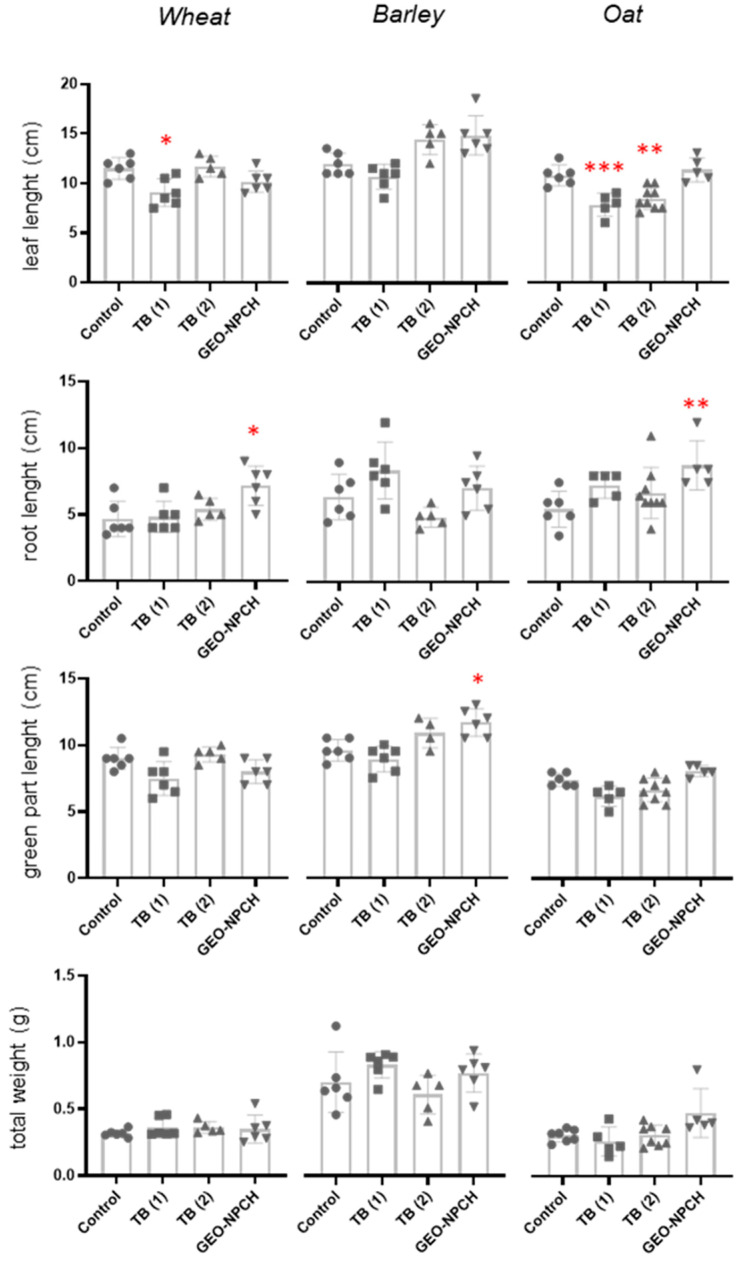
Gravimetric and longitudinal analyses at 30 days after the planting of treated plants. Control (plants without treatment; GEO-NPCH (seed plants treated with GEO-NPCH); TB1 (seed plants treated by spraying with TB); TB2 (seed plants supplied by the seed company). * *p* ≤ 0.05; ** *p* ≤ 0.01; *** *p* ≤ 0.001.

**Figure 5 polymers-16-03385-f005:**
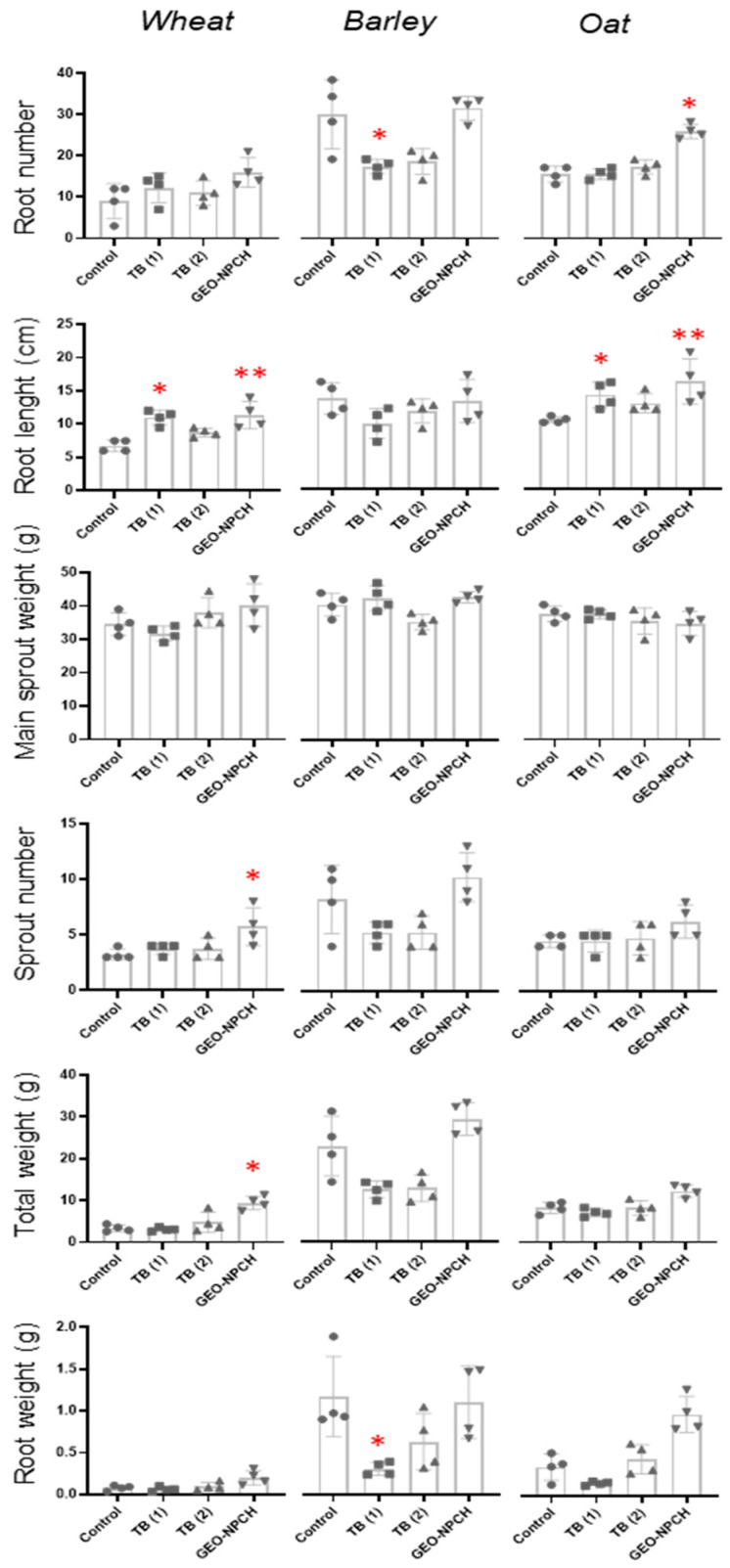
Gravimetric and longitudinal analyses at 90 days after planting of treated barley plants. Control (plants without tartar; GEO-NPCH (seed plants treated with GE-NPCH); TB1 (seed plants treated by spraying with TB); TB2 (seed plants supplied by the seed company). * *p* ≤ 0.05; ** *p* ≤ 0.01.

**Figure 6 polymers-16-03385-f006:**
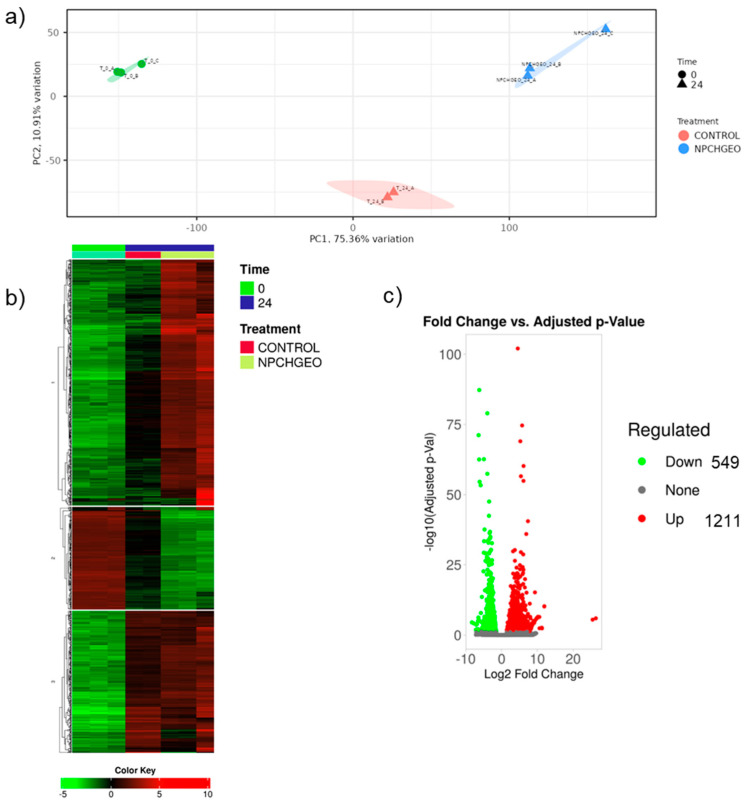
Overview of the DEG analysis among samples. (**a**) PCA plot of whole transcriptional data from seeds before and after 24 h of treatment with control or NPCHGEO. (**b**) Heatmap depicting the expression of the top 20 most variable genes among samples. (**c**) Volcano plot.

**Figure 7 polymers-16-03385-f007:**
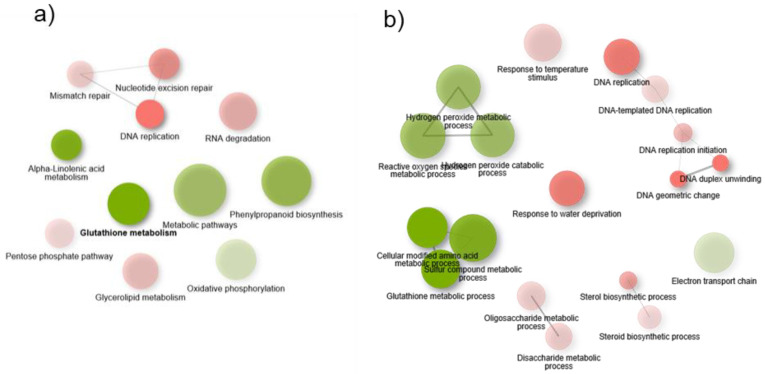
Network plot of main KEGG pathways (**a**) and a GO biological process (**b**) enrichment analysis using the GSEA (Pre-Ranked) method.

**Figure 8 polymers-16-03385-f008:**
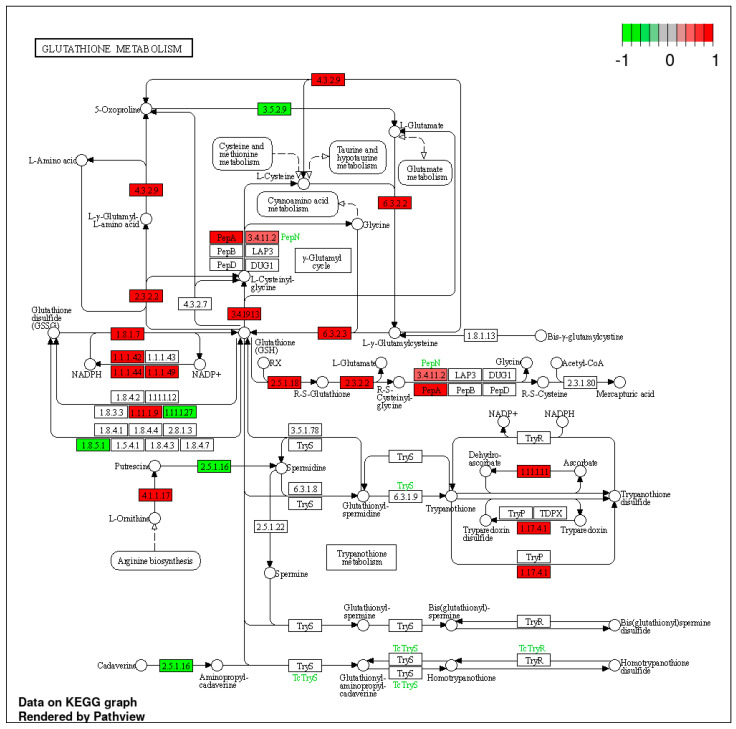
Glutathione metabolism pathway diagram. Colors indicate changes in expression according to the legend.

**Table 1 polymers-16-03385-t001:** Stability study of the GE-NPCH.

Time (Months)	Z-Ave (r.nm)	PDI	Z Potential (mV)
0	251.1 ± 3.3	0.3 ±0.0	+43.6 ± 0.1
2	263.2 ± 6.6	0.4 ± 0.0	+43.7 ± 0.0
5	299.9 ± 2.2	0.4 ± 0.0	+43.3 ± 0.1
10	309.9 ± 1.7	0.4 ± 0.0	+42.5 ± 0.2
11	339.0 ± 54.9	0.4 ± 0.1	+41.8 ± 0.5

**Table 2 polymers-16-03385-t002:** Total chlorophyll, polyphenols, and total flavonoids content in cereal plant samples collected after 90 days of cultivation.

Data Obtained from Plants 90 Days After Sowing			
Sample	Total Chlorophyll mg/g	Total Polyphenols mg/g	Total Flavonoids mg/g
WHEAT NT	8.54	11.63	13.15
WHEAT NP	8.24	8.66	11.88
WHEAT TB1	8.25	10.41	12.89
WHEAT TB2	8.46	9.38	12.49
BARLEY NT	6.08	8.83	11.21
BARLEY NP	3.48	12.88	12.86
BARLEY TB1	3.66	11.04	15.16
BARLEY TB2	3.38	14.21	11.70
OAT NT	3.01	2.07	8.57
OAT NP	4.21	3.54	10.35
OAT TB1	3.30	5.68	9.18
OAT TB2	4.44	3.82	9.80

## Data Availability

The data presented in this study are openly available in [RUIDERA] at [https://hdl.handle.net/10578/39177] accessed on 1 November 2024.

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
