# Peer review of "Field Crop Evaluation of Polymeric Nanoparticles of Garlic Extract–Chitosan as Biostimulant Seed Nano-Priming in Cereals and Transcriptomic Insights"

_polymers, 2024, doi:10.3390/polym16233385_

Round 1

Reviewer 1 Report

Comments and Suggestions for Authors

In this manuscript, the author evaluated the effect of eco-friendly garlic extract-chitosan on the seed growing, which is a meaningful work. However, the whole manuscript was in a bad organizing and writing, the following issues should be addressed before acceptance:

1. Line 70-76 shows the positive charged chitosan nanoparticles have been proved to regulate plants positive, why do you added garlic extract to chitosan nanoparticles (line 89-91)? what is your purpose? The properties of garlic extract should also discussed in the Introduction.

2. Line 93, what is “GE-NPCH”, which should defined for its first appeared.

3. Section 2.1, the supplier and composition of garlic extract should provided. In addition, the supplier and molecular weight of chitosan should also provided.

4. How can you obtained the GE-NPCH? Section 2.2 should be the preparation of GE-NPCH, the preparation process of the most important materials was missed.

5. Line 125, what is TB?

6. The label within Figure 3 is too vague, which should revised.

7. For Section 3.3. Evaluation of Seed Treatment with GE-NPCH in Rainfed Crop, the control experiments were missed, as both of garlic extract and chitosan nanoparticles may promote the growth of the plants, thus the effect of single garlic extract or chitosan nanoparticles should also investigated.

8. What is the effect and mechanism of garlic extract?

Author Response

Comments are in the pdf attached

Reviewer 2 Report

Comments and Suggestions for Authors

Undoubtedly, a very important aspect of modern agriculture is the ecological one. The development of new systems for accelerating seed germination, strengthening roots, accelerating fruit growth, that is, reasonable stimulation of metabolic processes underlying the growth, development and increase in the productivity of agricultural crops is a very important area of ​​modern science, including polymer chemistry. However, the developed stimulating systems must be non-toxic, biocompatible, biodegradable and, most importantly, environmentally friendly. This is a difficult and very important task. For an original and interesting solution to the above-mentioned problem, the authors developed a new phytosanitary product with biostimulating properties based on the encapsulation of garlic extract in a lower dose (<0.1%) in chitosan nanoparticles as a nano-priming agent for seeds. It was no coincidence that the authors used chitosan, since chitosan easily forms sedimentation and aggregation stable nanoparticles under the influence of external factors. In addition, chitosan is a natural biocompatible, biodegradable, non-toxic, environmentally friendly polymer. The resulting nanoparticles showed high stability when stored at 4 degrees Celsius. In addition, the obtained nanoparticles were active in stimulating the growth and development of wheat, barley and oats. It is very important that the authors not only recorded the fact of stimulation, but also identified the key mechanisms for the development of the stimulating effect (higher growth of genes for development, metabolism and plant response compared to untreated seeds). The article is written competently and logically, in good English. The authors refer to relevant articles when discussing the results. The authors widely use statistics and mathematical processing of experimental data. The findings correspond to the objectives of the work and the experimental results. The methodological part is well and competently written. I believe that this article should be published, but only after a major revision. This work is very good, but the impression of it is greatly spoiled by the abstract. It should be rewritten and should be understandable even to a non-specialist FROM THE FIRST WORDS. In the abstract, the first words should indicate that you have obtained an environmentally friendly GROWTH STIMULATOR for wheat, barley and oats. Then the authors must describe the production of nanoparticles (despite the fact that they were provided by another organization!!! this is very important for reproducibility). In addition, the characteristics of the chitosan used (molecular weight distribution, average molecular weight, degree of deacetylation and methods for determining these characteristics) must be provided. This is also of paramount importance for reproducibility. Finally, the authors must confirm that the systems obtained are truly environmentally safe. Obtaining nanoparticles from natural non-toxic compounds in no way guarantees their environmental safety.

Author Response

Comments are in the pdf attached

Round 2

Reviewer 1 Report

Comments and Suggestions for Authors

All of the issues mentioned were resolved in detail

Reviewer 2 Report

Comments and Suggestions for Authors

The authors have improved the manuscript. Of course, this article will be interesting to many readers and I recommend accepting it.